# Differential occupational risks to healthcare workers from SARS-CoV-2 observed during a prospective observational study

David W Eyre[1,2,3,4]*, Sheila F Lumley[2], Denise O'Donnell[5], Mark Campbell[2], Elizabeth Sims[2], Elaine Lawson[2], Fiona Warren[2], Tim James[2], Stuart Cox[2], Alison Howarth[5], George Doherty[5], Stephanie B Hatch[5,6], James Kavanagh[5], Kevin K Chau[5], Philip W Fowler[3,5], Jeremy Swann[5], Denis Volk[3,5], Fan Yang-Turner[3,5], Nicole Stoesser[2,3,4,5], Philippa C Matthews[2,5], Maria Dudareva[2], Timothy Davies[2], Robert H Shaw[2], Leon Peto[2], Louise O Downs[2], Alexander Vogt[2], Ali Amini[2,5], Bernadette C Young[2,5], Philip George Drennan[2], Alexander J Mentzer[2,5], Donal T Skelly[2,7], Fredrik Karpe[3,8], Matt J Neville[3,8], Monique Andersson[2], Andrew J Brent[2], Nicola Jones[2], Lucas Martins Ferreira[5], Thomas Christott[5], Brian D Marsden[5,9], Sarah Hoosdally[3,5], Richard Cornall[5], Derrick W Crook[2,3,5], David I Stuart[5], Gavin Screaton[5], Oxford University Hospitals Staff Testing Group, Timothy EA Peto[2,3,5], Bruno Holthof[2], Anne-Marie O'Donnell[2], Daniel Ebner[5,6], Christopher P Conlon[2,5], Katie Jeffery[2†], Timothy M Walker[2,5,10†]*

*For correspondence:
david.eyre@bdi.ox.ac.uk (DWE);
timothy.walker@ndm.ox.ac.uk
(TMW)

†These authors contributed
equally to this work

Group author details:
Oxford University Hospitals Staff
Testing Group See page 16

[1]Big Data Institute, Nuffield Department of Population Health, University of Oxford, Oxford, United Kingdom; [2]Oxford University Hospitals NHS Foundation Trust, Oxford, United Kingdom; [3]NIHR Oxford Biomedical Research Centre, University of Oxford, Oxford, United Kingdom; [4]NIHR Health Protection Research Unit in Healthcare Associated Infections and Antimicrobial Resistance at University of Oxford in partnership with Public Health England, Oxford, United Kingdom; [5]Nuffield Department of Medicine, University of Oxford, Oxford, United Kingdom; [6]Target Discovery Institute, University of Oxford, Oxford, United Kingdom; [7]Nuffield Department of Clinical Neurosciences, University of Oxford, Oxford, United Kingdom; [8]Radcliffe Department of Medicine, University of Oxford, Oxford, United Kingdom; [9]Kennedy Institute of Rheumatology Research, University of Oxford, Oxford, United Kingdom; [10]Oxford University Clinical Research Unit, Ho Chi Minh City, Viet Nam

Competing interest: See
page 17

Reviewing editor: Marc
Lipsitch, Harvard TH Chan
School of Public Health, United
States

**Abstract** We conducted voluntary Covid-19 testing programmes for symptomatic and asymptomatic staff at a UK teaching hospital using naso-/oro-pharyngeal PCR testing and immunoassays for IgG antibodies. 1128/10,034 (11.2%) staff had evidence of Covid-19 at some time. Using questionnaire data provided on potential risk-factors, staff with a confirmed household contact were at greatest risk (adjusted odds ratio [aOR] 4.82 [95%CI 3.45–6.72]). Higher rates of Covid-19 were seen in staff working in Covid-19-facing areas (22.6% vs. 8.6% elsewhere) (aOR 2.47 [1.99–3.08]). Controlling for Covid-19-facing status, risks were heterogenous across the hospital, with higher rates in acute medicine (1.52 [1.07–2.16]) and sporadic outbreaks in areas with few or no Covid-19 patients. Covid-19 intensive care unit staff were relatively protected (0.44 [0.28–0.69]), likely by a bundle of PPE-related measures. Positive results were more likely in Black (1.66 [1.25–

2.21]) and Asian (1.51 [1.28–1.77]) staff, independent of role or working location, and in porters and cleaners (2.06 [1.34–3.15]).

## Introduction

On 23rd March 2020 the UK followed other European countries in locking down its population to mitigate the impact of the rapidly evolving Covid-19 pandemic. By 5th May the UK had recorded Europe's highest attributed death toll (*Johns Hopkins Coronavirus Resource Centre, 2020*).

Lock-down isolated many UK households but staff maintaining healthcare services continued to be exposed to patients and to other healthcare workers (HCW). National Health Service (NHS) hospitals endeavoured to provide personal protective equipment (PPE) in line with Public Health England (PHE) guidelines in clinical areas and encouraged social distancing elsewhere. Despite these measures the incidence of Covid-19 among HCWs is higher than in the general population (*Nguyen et al., 2020*; *Disparities in the risk and outcomes from COVID-19, 2020*).

Multiple studies have investigated Covid-19 in HCWs (*Nguyen et al., 2020*; *Rivett et al., 2020*; *Shields et al., 2020*; *Houlihan et al., 2020*). However, crucial to designing a safe working environment and maintaining effective healthcare services is an understanding of the risks associated with specific roles and to individuals, and whether risk is associated with social-mixing, direct exposure to Covid-19 patients or PPE type. Some studies have suggested exposure to Covid-19 patients poses increased risk (*Nguyen et al., 2020*; *Ran et al., 2020*; *Lombardi et al., 2020*), whilst others have not (*Hunter et al., 2020*; *Galan et al., 2020*; *Folgueira et al., 2020*). However, none have addressed these questions by comprehensively investigating all staff groups across an institution, simultaneously assessing symptomatic and asymptomatic incidence.

Alongside routine SARS-CoV-2 PCR testing of symptomatic staff, Oxford University Hospitals NHS Foundation Trust (OUH) has offered SARS-CoV-2 PCR and antibody testing to all asymptomatic staff to improve infection prevention and control for staff and patients. We present the results of this large, high-uptake programme.

## Results

### Oxford University Hospitals Covid-19 context

From mid-March 2020 OUH saw daily admissions of patients with Covid-19. By 8th June, 636 patients had been admitted within a week of a confirmed Covid-19 diagnosis. Weekly incidence of new Covid-19 diagnoses in these patients peaked during the week beginning 30th March (n = 136/week, *Figure 1A*). Routine SARS-CoV-2 PCR testing of symptomatic staff (with fever or new persistent cough) began on 27th March; weekly incidence of new staff diagnoses peaked the week beginning 6th April (n = 98/week, *Figure 1B*). Up to and including the 8th June, 348/1498 (23%) symptomatic staff tested were PCR-positive (2.5% of all 13,800 staff employed at OUH). Ten staff were admitted to hospital with Covid-19 (0.07%); four died (0.03%).

### Asymptomatic staff testing

A voluntary asymptomatic screening programme offering SARS-CoV-2 PCR and antibody testing to all staff working anywhere on site commenced on 23rd April 2020. Between 23rd April and 8th June, 10,610 of the 13,800 (77%) staff employed by OUH registered for asymptomatic testing and 10,034 (73%) were tested at least once, 9926 by PCR and 9958 by serology. The majority of testing was undertaken in the first three weeks of May 2020 (*Figure 1C*). 288/9926 (2.9%) staff were PCR-positive on their first asymptomatic screen; 145 were permitted to remain at work: 61 (21%) had tested PCR-positive >7 days previously while symptomatic and had since recovered and 84 (29%) had a history suggestive of previous Covid-19 (in most, prior to the availability of symptomatic staff testing). The remainder, 130/288 (45%), were assessed to have a new infection and self-isolated. Documentation was incomplete for six staff and seven could not be contacted.

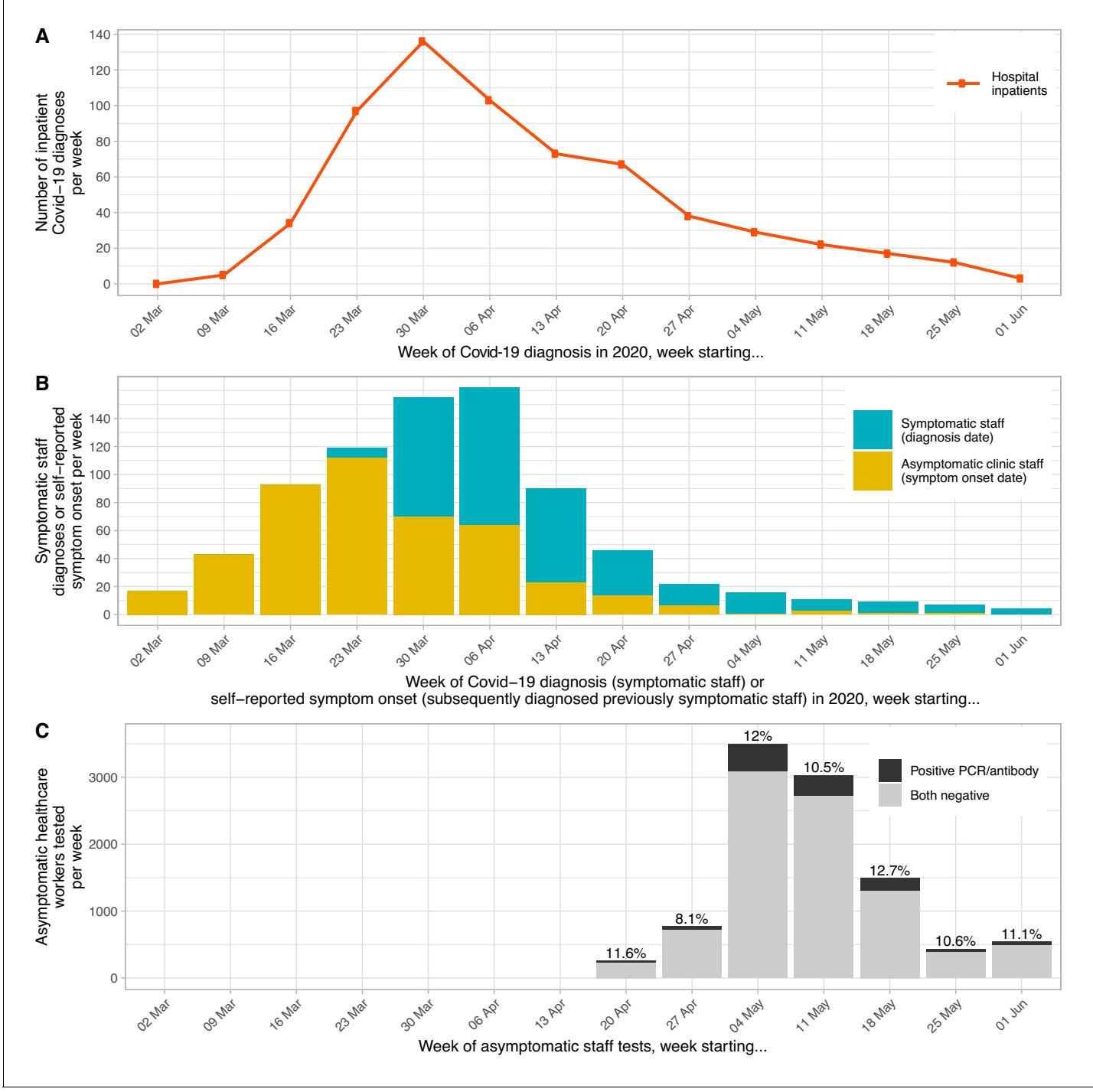

**Figure 1.** Epidemiological curve for hospital inpatients (panel A) and staff (panel B) diagnosed with Covid-19, by week and timing of asymptomatic staff testing (panel C). Each patient admitted to hospital with a diagnosis of Covid-19 within ±7 days of any day during their admission is plotted based on the date of their positive PCR test. Testing for symptomatic staff was made available from 27th March 2020; staff were asked to attend on days 2–4 of symptoms and are plotted in the week of their positive test. Of 1128 staff positive by PCR or serology at the asymptomatic staff clinic, 192 had been previously diagnosed at the symptomatic staff clinic. Of the remaining 936 positive staff, 449 (48%) reported a date when they believed a Covid-19 illness had begun, these are plotted in yellow above, many with symptoms before the availability of staff testing. As 487 (52%) of staff did not provide a date of symptom onset the true values for the yellow bars on the y-axis are likely to be around two times higher. Panel C shows the week asymptomatic staff were tested, those testing SARS-CoV-2 PCR-positive and/or IgG-positive are shown in black and those with negative tests in grey. The overall percentage of staff tested each week with positive PCR and/or antibody results is shown above each bar. The bar for 01 June also includes 31 staff tested on 08 June.

## Duration of PCR positivity

Having observed asymptomatic staff who were PCR-positive following symptomatic recovery, we investigated the duration of PCR positivity using data from staff and patients with consecutive tests. Repeat testing of patients was guided by individual clinician request, in conjunction with the infection consult service. Repeat testing of staff was available in those attending asymptomatic screening who had previously been tested by the symptomatic testing service and was also undertaken up to weekly in the cohort of staff who attended the asymptomatic testing service during the first week of testing. Fewer staff than patients were persistently positive at 7–13 days (exact p=0.003), but results were similar by 14–20 days, 68/159 (43% [95% CI 35–51%]) overall. 34/141 (24% [17–32%]) samples taken after ≥42 days were positive (*Figure 2*).

## Combined serology and PCR results in asymptomatic staff

Considering the first asymptomatic clinic PCR and serology samples from each staff member, 1128/10,034 (11.2%) staff attending for asymptomatic screening were positive by PCR or serology, indicating a composite primary outcome of 'Covid-19 at some time', including 192 previously diagnosed via symptomatic staff testing. 1069/9958 (10.7%) staff with an immunoassay result were IgG-positive (see *Supplementary file 1A* for a comparison of results by the two assays). In staff providing

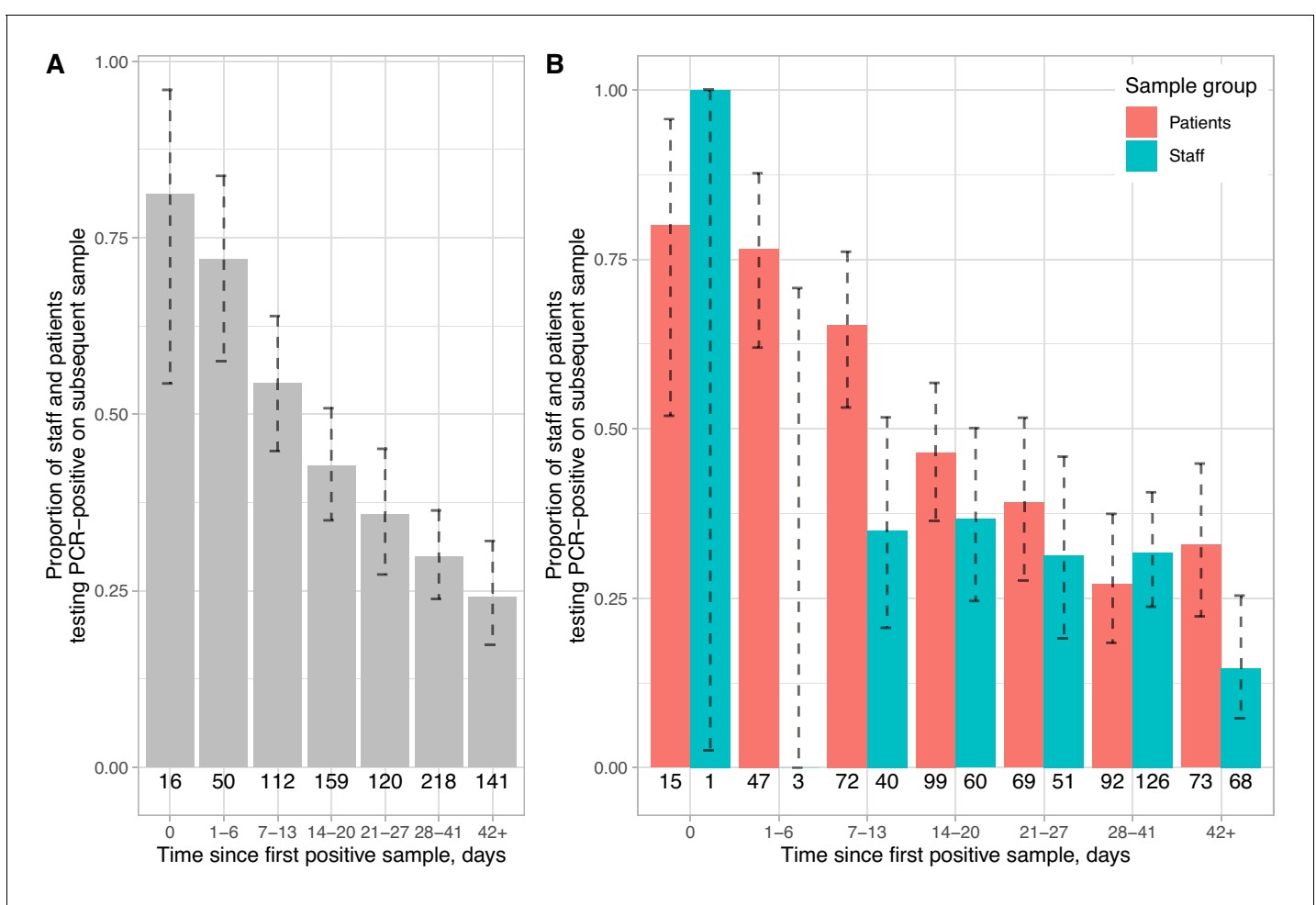

**Figure 2.** Proportion of staff and patients remaining PCR-positive on repeat nasopharyngeal swabs. Panel A shows pooled data and Panel B data separately for staff and patients. The number of individuals with a repeat test in each time interval is shown below each bar and 95% exact binomial confidence intervals are plotted. All tests following a first positive sample are included up until the first negative sample per patient. The number of tests positive after a repeat swab on the same day is indicative of the sensitivity of a single swab, 15/16 of these swabs were obtained from patients on wards by any available staff member, whereas staff sampling was undertaken by specially trained teams.

questionnaire data prior to asymptomatic testing, 552/1126 (49.0%) staff subsequently testing positive thought they had already had Covid-19, compared to 1106/8906 (12.4%) testing negative.

## Symptoms predictive of Covid-19

We asked all staff attending asymptomatic screening about possible Covid-19-related symptoms since 1st February 2020 (*Table 1*). In a multivariable model containing all symptoms, anosmia or loss of taste was most strongly predictive of Covid-19 (aOR 17.7 [95%CI 14.1–22.2], p<0.001). Other independent predictors included myalgia, fever and cough. Adjusting for other symptoms, sore throat was a negative predictor for Covid-19.

## Risk factors for Covid-19 in healthcare workers

We used pre-test questionnaire data provided by 10,032 asymptomatic staff to estimate risk factors for Covid-19 (two staff tested did not provide questionnaire data). Staff diagnosed via the symptomatic testing clinic alone were not included as no detailed questionnaire data were collected from these staff. However, the 192/348 (55%) staff diagnosed by the symptomatic testing service who subsequently attended the asymptomatic clinic were included.

67/174 (38.5%) staff reporting household contact with a PCR-confirmed case tested positive, compared to 1059/9858 (10.7%) without (p<0.001). SARS-CoV-2 infected staff were also more likely to report suspected, but unconfirmed contacts, and non-household contacts (*Figure 3*, *Supplementary file 1B*). 368/2165 (17.0%) staff reporting workplace contact without PPE with a known or suspected Covid-19 patient tested positive, compared with 758/7867 (9.6%) not reporting similar exposure (p<0.001). To mitigate recall bias, we repeated this analysis restricted to staff who

**Table 1.** Association of self-reported symptoms and Covid-19 in hospital staff.

| Symptom | Symptom reported | | | | Symptom not reported | | | | Univariable | | Multivariable | |
|---|---|---|---|---|---|---|---|---|---|---|---|---|
| | n | Covid-19 positive | Covid-19 negative | % positive | N | Covid-19 positive | Covid-19 negative | % positive | Or (95% CI) | P value | Or (95% CI) | P value |
| Anosmia or loss of taste | 858 | 489 | 369 | 57.0 | 9174 | 637 | 8537 | 6.9 | 17.7 (15.1–20.8) | <0.001 | 17.7 (14.1–22.2) | <0.001 |
| Myalgia | 1796 | 501 | 1295 | 27.9 | 8236 | 625 | 7611 | 7.6 | 4.7 (4.1–5.4) | <0.001 | 2.1 (1.7–2.6) | <0.001 |
| Fever | 1465 | 406 | 1059 | 27.7 | 8567 | 720 | 7847 | 8.4 | 4.2 (3.6–4.8) | <0.001 | 1.5 (1.2–1.8) | <0.001 |
| Nausea or vomiting | 417 | 130 | 287 | 31.2 | 9615 | 996 | 8619 | 10.4 | 3.9 (3.1–4.9) | <0.001 | 1.2 (0.9–1.6) | 0.18 |
| Fatigue | 2718 | 591 | 2127 | 21.7 | 7314 | 535 | 6779 | 7.3 | 3.5 (3.1–4) | <0.001 | 1.0 (0.8–1.2) | 0.81 |
| Cough | 1813 | 403 | 1410 | 22.2 | 8219 | 723 | 7496 | 8.8 | 3 (2.6–3.4) | <0.001 | 1.2 (1.0–1.5) | 0.04 |
| Shortness of breath | 1022 | 245 | 777 | 24.0 | 9010 | 881 | 8129 | 9.8 | 2.9 (2.5–3.4) | <0.001 | 1.2 (0.9–1.5) | 0.30 |
| Diarrhoea | 607 | 147 | 460 | 24.2 | 9425 | 979 | 8446 | 10.4 | 2.8 (2.2–3.4) | <0.001 | 1.1 (0.9–1.5) | 0.30 |
| Hoarseness | 645 | 136 | 509 | 21.1 | 9387 | 990 | 8397 | 10.5 | 2.3 (1.8–2.8) | <0.001 | 1.2 (0.9–1.7) | 0.23 |
| Nasal congestion | 1871 | 355 | 1516 | 19.0 | 8161 | 771 | 7390 | 9.4 | 2.2 (2–2.6) | <0.001 | 1.0 (0.8–1.2) | 0.63 |
| Sore throat | 2248 | 356 | 1892 | 15.8 | 7784 | 770 | 7014 | 9.9 | 1.7 (1.5–2) | <0.001 | 0.6 (0.5–0.8) | <0.001 |
| *Hoarseness + Anosmia or loss of taste | | | | | | | | | | | 0.5 (0.3–0.8) | 0.002 |
| *Shortness of breath + Anosmia or loss of taste | | | | | | | | | | | 0.5 (0.3–0.7) | <0.001 |

*All interactions with an interaction Wald p values < 0.01 are shown.

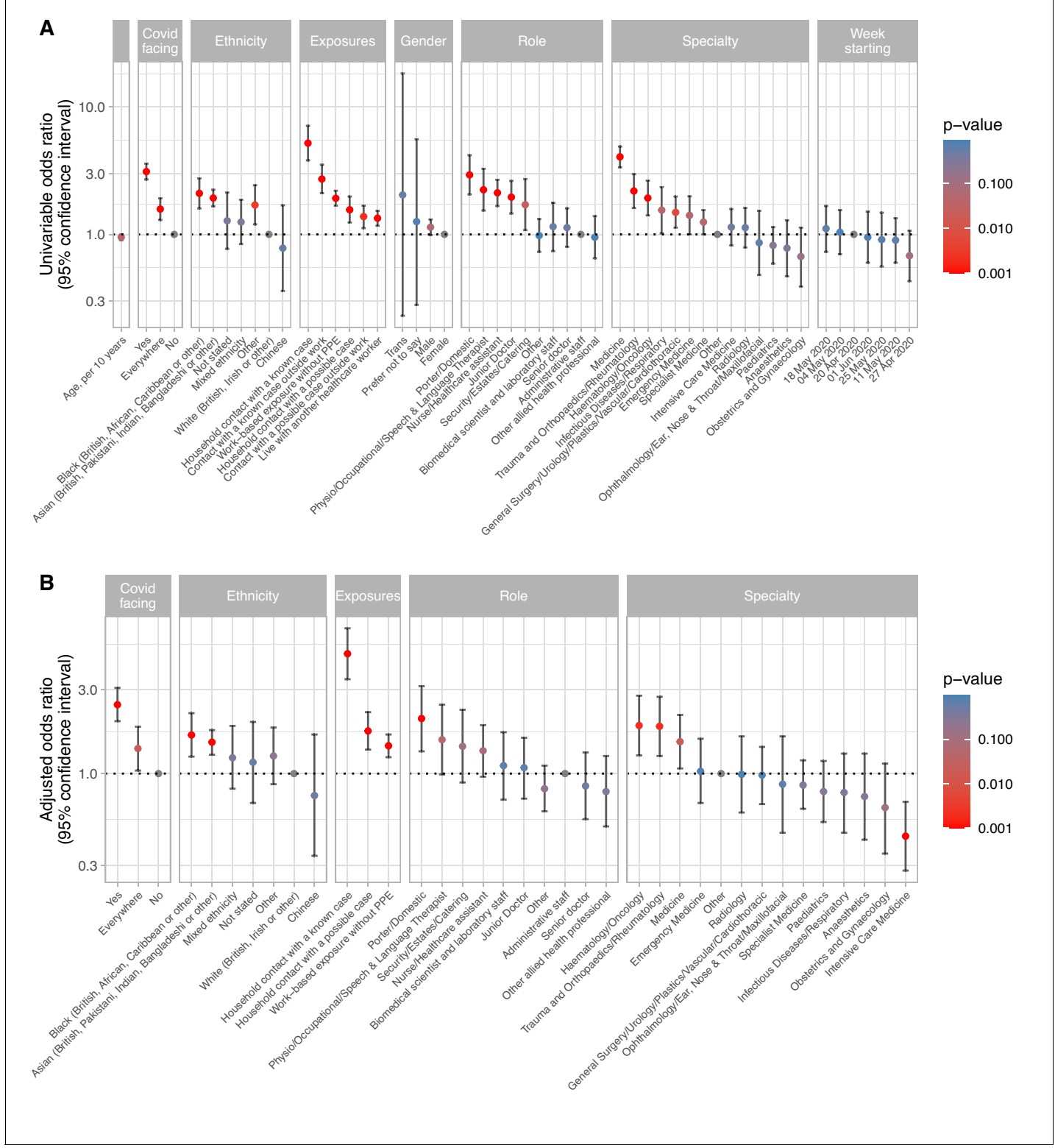

**Figure 3.** Univariable (panel A) and multivariable (panel B) relationships between risk factors and staff infection with SARS-CoV-2 in 10,032 healthcare workers. See *Supplementary file 1B* for count data, univariable and multivariable odds ratios. Pairwise interactions were sought between all variables the multivariable model, a single interaction exceeded the p<0.01 screening threshold, representing decreased risk of Covid-19 in emergency department staff reporting exposure to a Covid-19 without PPE (p=0.002). However, given the large number of interactions sought and biological

*Figure 3 continued on next page*

*Figure 3 continued*

implausibility, the interaction is omitted from the model presented. For the purpose of plotting p values <0.001 were rounded up to 0.001. Risk factor data were not available for two staff members. In panel A, the category for 01 June also includes 31 staff tested on 08 June.

did not think they had had Covid-19: 167/1653 (10.1%) reporting an exposure were positive compared to 407/6721 (6.1%) who did not (p<0.001).

We further investigated risk of workplace Covid-19 acquisition. 358/1586 (22.6%) staff on wards caring for patients with Covid-19 were infected, compared to 631/7369 (8.6%) on non-Covid-19 facing wards/other areas, and 139/1079 (12.9%) staff working across multiple areas (p<0.001). Covid-19 facing areas included the emergency department, acute medical and surgical wards, the respiratory high dependency unit (HDU) and three intensive care units (ICUs). However, the proportion of staff with a positive test working in acute medicine (222/793, 28.0%) was greater than in the emergency department (41/344, 11.9%) and in the ICUs (44/448, 9.8%) (*Figure 3A*, *Figure 4*, *Supplementary file 1B*).

Rates of Covid-19 infection varied by staff occupational role: porters and cleaners had the highest rates (60/323, 18.6%), followed by physio-, occupational and speech and language therapists (47/316, 14.9%) and nurses/healthcare-assistants (562/3971, 14.2%). Junior medical staff had higher

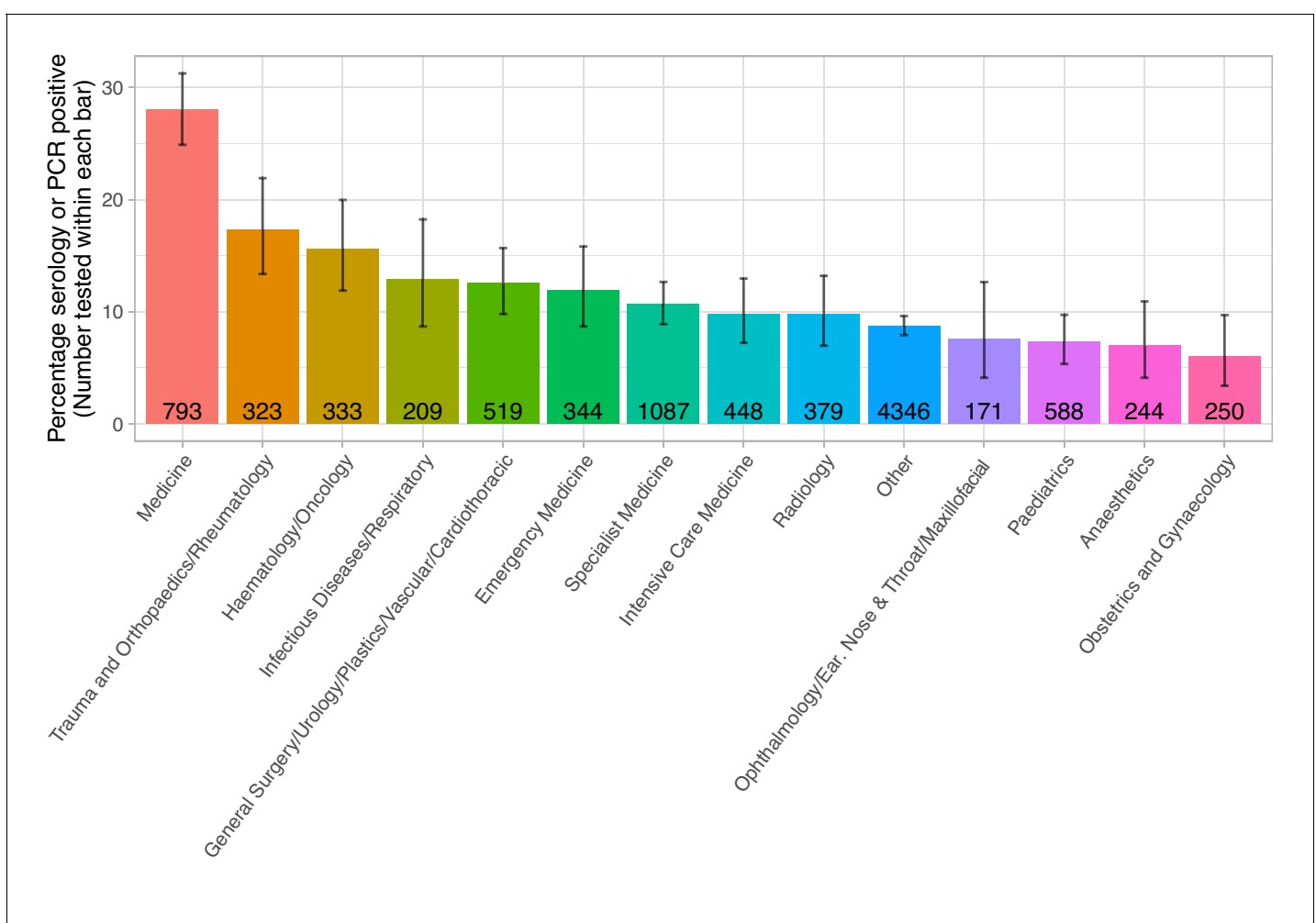

**Figure 4.** Proportion of staff testing positive by specialty area. The number of staff tested within each speciality is shown within each bar. The error bar indicates the 95% confidence interval. The 'Other' group includes staff members without a self-reported specialty. Staff working in a specialty area are predominantly nurses, healthcare assistances, doctors and therapists.

rates (113/853, 13.2%) than senior medical staff (57/704, 8.1%). Administrative staff had the lowest proportion (88/1218, 7.2%) of any major staff group (*Figure 3A*, *Figure 5*, *Supplementary file 1B*).

There was limited evidence that male staff were more at risk of infection than female staff (313/2562 [12.2%] positive vs. 812/7452 [10.9%], p=0.07) and that risk decreased with increasing age (univariable odds ratio [OR], per 10 years, 0.95 [95%CI 0.90–1.00, p=0.04], *Figure 6*). Covid-19 rates varied by self-described ethnicity. 686/7237 (9.5%) staff describing themselves as White (British/Irish/other) were infected, compared to 281/1673 (16.8%) and 71/394 (18.0%) staff describing themselves as Asian (British/Pakistani/Indian/Bangladeshi/other) or Black (British/African/Caribbean/other) respectively. Rates in staff describing themselves of mixed ethnicity or Chinese were 28/242 (11.6%) and 7/93 (7.5%) (*Figure 3A*, *Figure 7*, *Supplementary file 1B*). There was no evidence that the proportion of asymptomatic staff with a positive PCR and/or antibody varied by week of testing, in keeping with most asymptomatic staff testing occurring after the peak in Covid-19 in the hospital (*Figure 1A–C*).

## Risk factors: multivariable analysis

In multivariable analysis (*Figure 3B*, *Supplementary file 1B*), controlling for factors including hospital-based Covid-19 exposure, role, specialty and ethnicity, household contact with known (adjusted OR [aOR] 4.82, 95% CI 3.45–6.72, p<0.001) or suspected (1.75, 1.37–2.24, p<0.001) cases remained important risk factors. Working in Covid-19 facing areas (2.47, 1.99–3.08, p<0.001) or throughout the hospital (1.39, 1.04–1.85, p=0.02) was associated with increased risk compared to non-Covid-19 areas, as was workplace-based exposure to a suspected or known Covid-19-positive patient without PPE (1.44, 1.24–1.67, p<0.001). The latter could not be entirely accounted for by recall-bias as the association persisted restricting to staff who did not think they had had Covid-19 (1.30, 1.06–1.59, p=0.01).

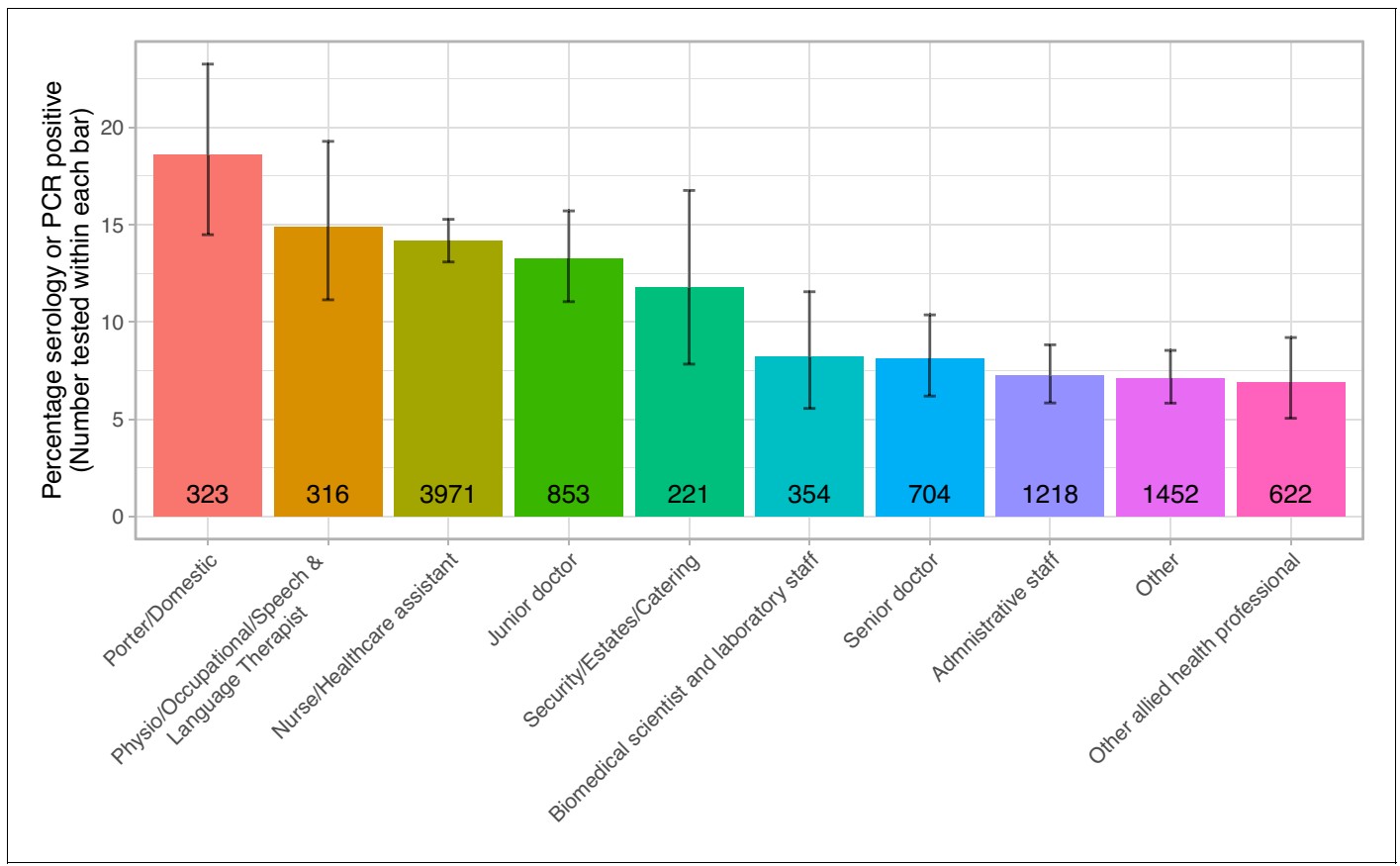

**Figure 5.** Proportion of staff testing positive by role. The number of staff tested within each role is shown within each bar. The error bar indicates the 95% confidence interval.

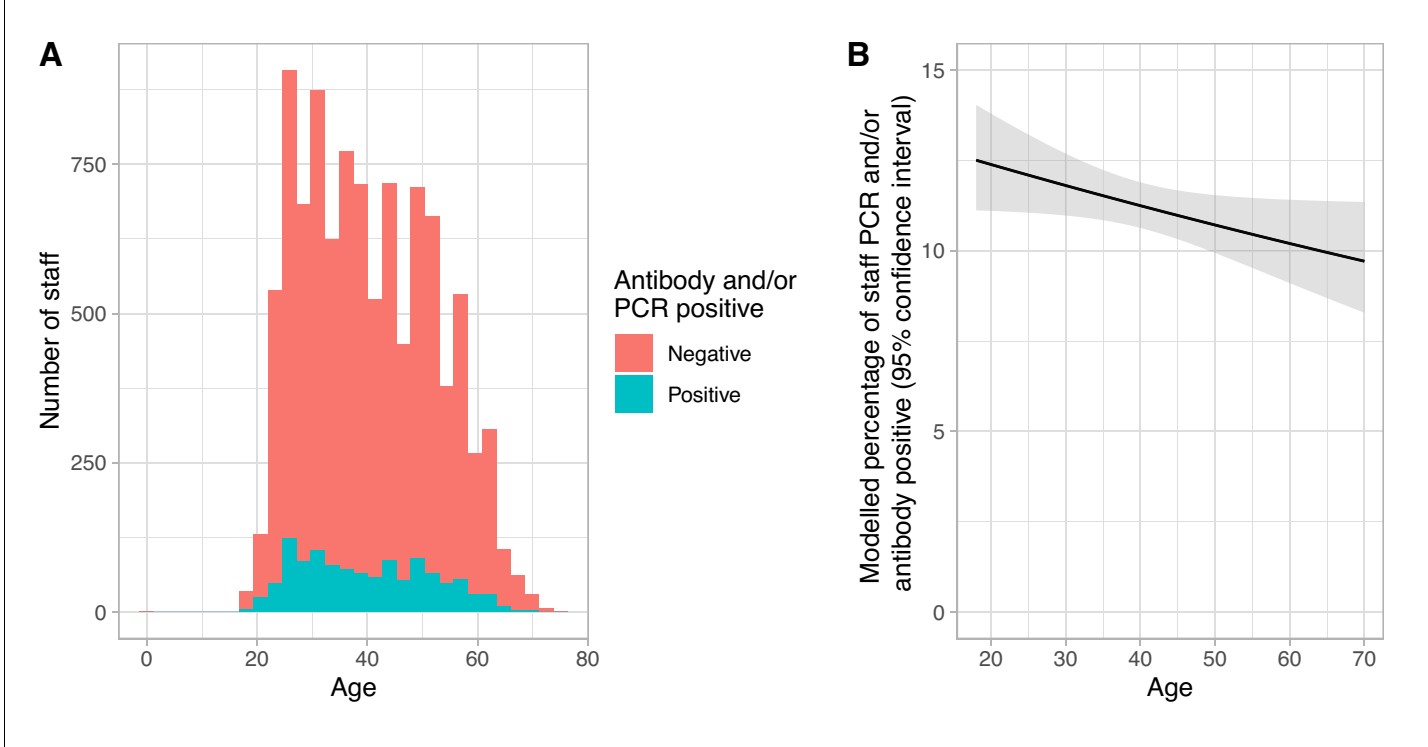

**Figure 6.** Relationship between age and Covid-19 infection in hospital staff. Panel A shows a histogram of staff ages for those attending asymptomatic screening, staff with a positive SARS-CoV-2 IgG antibody and/or PCR test at their first asymptomatic clinic attendance as shown in blue and those who were both PCR and antibody negative are shown in red. Panel B shows the univariable modelled percentage of staff positive by age, the solid line shows the expected value and the ribbon the 95% confidence interval.

Risk of Covid-19 infection varied by speciality, even after accounting for working in a Covid-19 facing area. Those working in acute medicine were at increased risk (aOR 1.52, 95% CI 1.07–2.16, p=0.02), while those working in ICUs were at lower risk (0.44, 0.28–0.69, p<0.001). Increased risk was also seen in in orthopaedics and haematology, reflecting staff-based outbreaks as these wards saw very few Covid-19 patients. The greatest risk of infection by role remained for porters and cleaners (2.06, 1.34–3.15, p=0.001). By ethnic group, Black (1.66, 1.25–2.21, p<0.001) and Asian (1.51, 1.28–1.77, p<0.001) staff were at greatest risk of Covid-19.

Risk factors for presence of SARS-CoV-2 IgG antibodies were very similar to the main model with a composite point including PCR results. The same factors were selected in the multivariable model (*Supplementary file 1C*), with the addition of gender: male healthcare workers had increased risk of SARS-CoV-2 seropositivity (aOR 1.19, 95% CI 1.01–1.40, p=0.03).

## Heterogeneity in risk of Covid-19 in healthcare workers between hospitals and wards

We investigated the relationship between infectious pressure from patients and the proportion of staff infected by considering each admitted patient infectious from −2 to +7 days around their first positive SARS-CoV-2 PCR. At a hospital building level (*Figure 8A*), the two buildings admitting most patients with Covid-19 had higher levels of staff infection (14.1%, 15.3%) than the majority of other buildings (5.4–8.6%). However, one site with low rates of patient infection and another, non-clinical site without patients had rates of 13.5% and 19.7% respectively. At a ward level (*Figure 8B*), there was only a weak positive correlation between Covid-19 pressure from patients and staff infection rates ($R^2$ = 0.09, p=0.02). ICUs and the HDU had lower rates of staff infection for a given Covid-19 pressure than general Covid-19 facing wards (adjusted linear regression coefficient −29% [95% CI −46%, −12%; p=0.002]). While dedicated Covid-19 cohort wards had similar rates of staff Covid-19 to general wards overall (*Supplementary file 1D*), several general wards had much higher rates (*Figure 8B*).

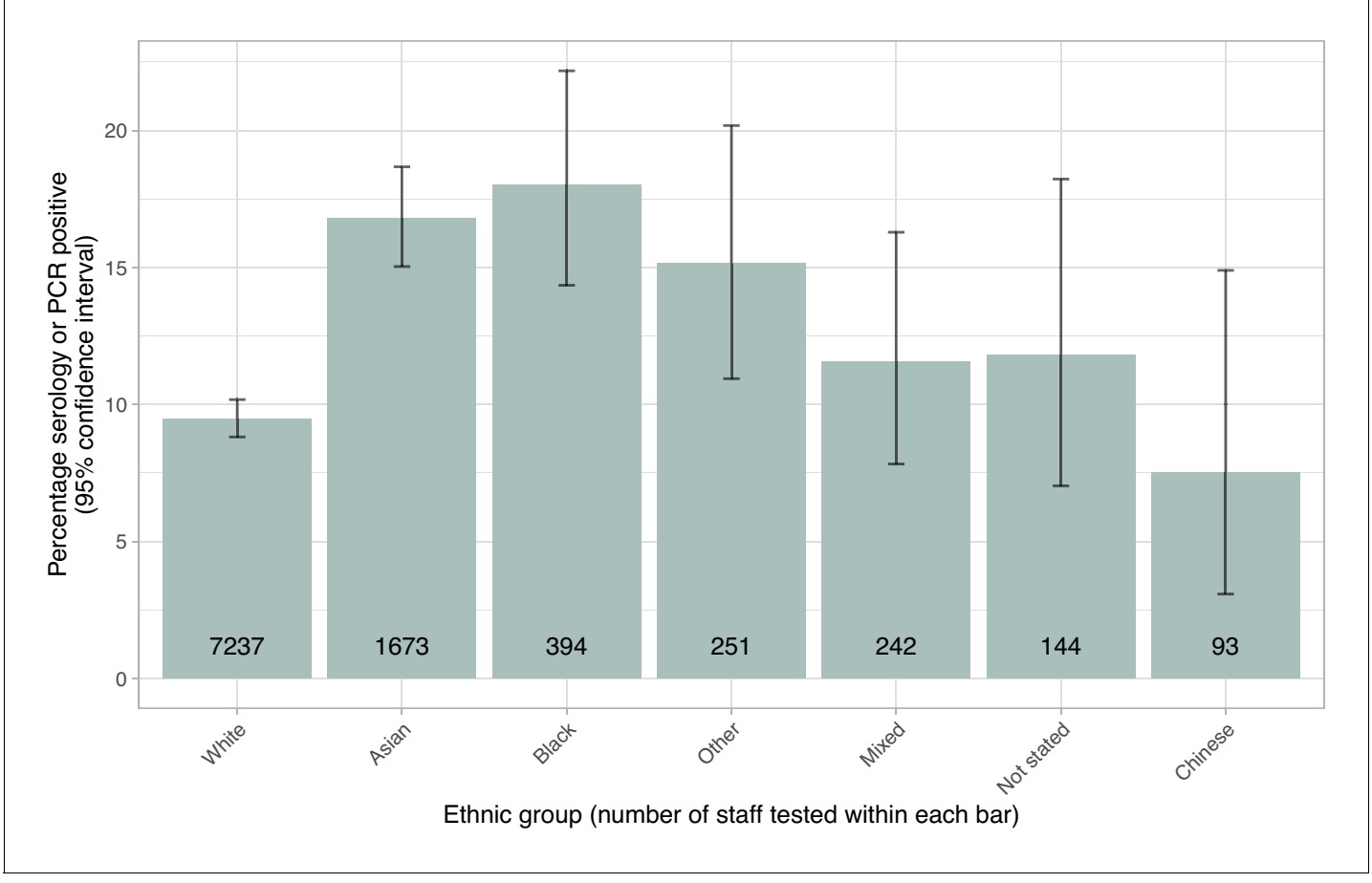

**Figure 7.** Proportion of staff testing positive by self-described ethnicity. The number of staff tested within each group is shown within each bar. The error bar indicates the 95% confidence interval.

### Contact tracing

PCR-positive asymptomatic staff who had not previously had Covid-19 were asked to name all colleagues with whom they had had >5 min of face-to-face conversation or been within 2 m for >15 min, within the past 48 hr, without a face mask. During the first 2 weeks of asymptomatic screening, 130 contacts were tested 7 days after contact with their index case, and 62 re-attended at day 14. Only one contact tested positive. As this rate of detection was below the background rate, contact tracing was discontinued for asymptomatic staff.

## Discussion

We present the results of a large and comprehensive Covid-19 staff testing programme across four teaching hospital sites in one UK county, attended by 73% of 13,800 staff employed by OUH. Using a composite outcome of either a positive PCR or serology result, by 8th June we detected evidence of Covid-19 at some time in 11.2% of staff. Put in context, UK-wide seroprevalence was 6.8% on 28th May 2020, with a higher incidence among healthcare workers than in the general population (*Office of National Statistics Coronavirus, 2020*).

We observed varying risk to our hospital staff associated with working location, occupational role and demographic factors. The greatest risk was associated with Covid-19 infected household contacts (although only 38.5% of staff with a contact became infected) and with working in Covid-19-facing areas (22.6% vs. 8.6% elsewhere) where there was one additional SARS-CoV-2 infection per ~7 staff compared to elsewhere. On univariable analysis staff with most direct patient contact were at increased risk including porters, cleaners, nurses, healthcare-assistants, therapists and junior doctors.

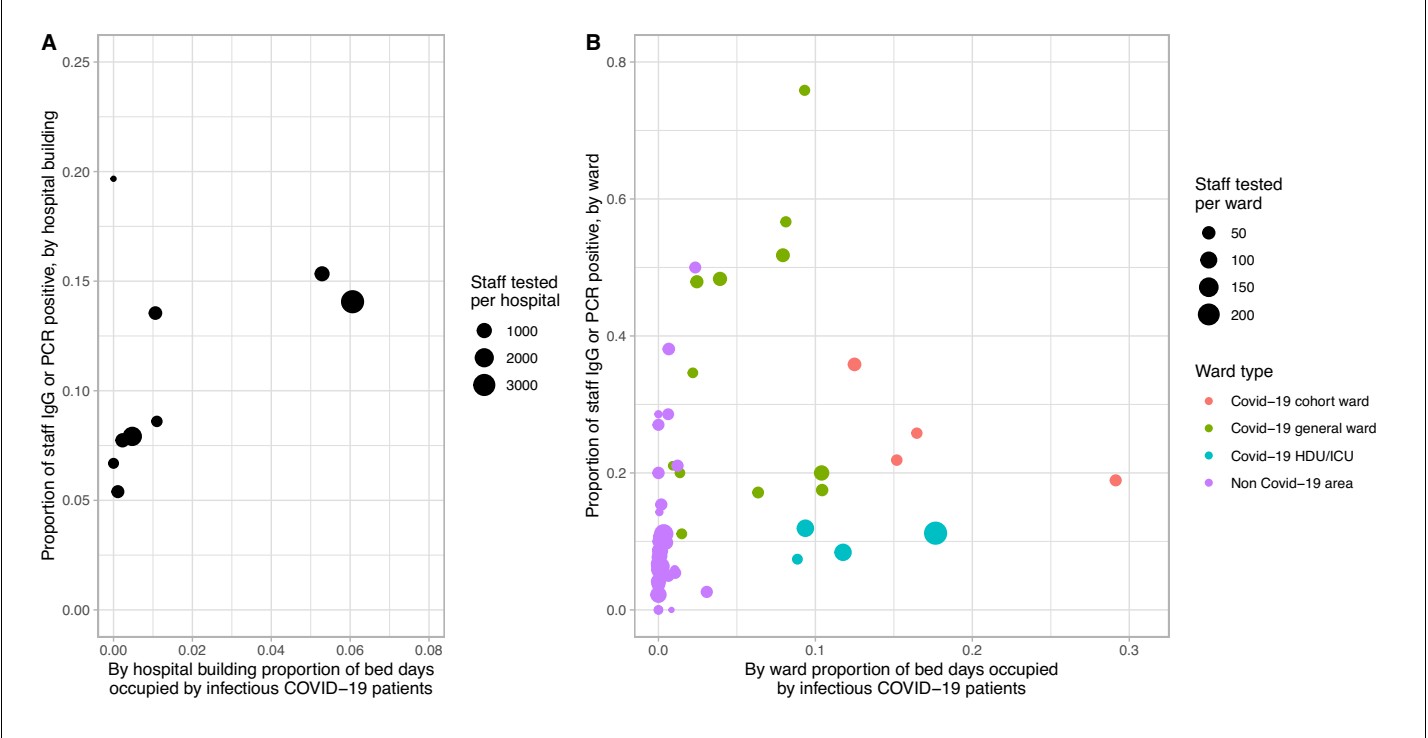

**Figure 8.** Proportion of staff infected by extent of Covid-19 infectious pressure from patients, by eight hospital buildings across four hospitals (panel A) and by ward (panel B). Covid-19 infectious pressure was calculated by considering each patient infectious from −2 to +7 days around the date of their first positive SARS-CoV-2 PCR test. Only staff working in a single hospital or ward are included in the plot. Wards with fewer than 10 staff tested are not plotted. Covid-19 cohort wards admitted only patients with suspected or known Covid-19, whereas Covid-19 general wards were acute medical wards receiving new admissions and acute medical patients initially believed not to have Covid-19. Non Covid-19 areas did not admit suspected Covid-19 patients and any suspected or confirmed Covid-19 patients were transferred off these wards as soon as possible.

Adjusting for working in a Covid-19 area captured much of this risk, except for porters and cleaners who had the highest adjusted risk of any staff group, and who typically operate across the hospital.

A heterogenous pattern also emerged across different Covid-19-facing areas. Risk seen on acute medical wards was greater than in the emergency department which was often bypassed by Covid-19 patients, whilst working on a Covid-19 facing ICU was relatively protective. One key difference across these areas was the type of PPE worn and the time periods over which it was mandated. Level-2 PPE (full length gown, gloves, correctly fitted FFP3/N99 mask and full face visor) was mandatory on ICU and HDU throughout, whereas policies changed over time on other wards (*Table 2*). Moreover, staff on ICU and HDU received extensive training in donning and doffing and had dedicated space and supervision for this whereas ward staff did not. Prior to 1st April 2020, in line with national guidance, in acute medical areas outside of Covid-19 cohort wards level-1 PPE (fluid resistant surgical mask, gloves, apron and optional eye protection) was only worn for contact with patients with known or suspected Covid-19, potentially leading to unprotected exposure to patients in whom Covid-19 was not suspected, such as afebrile elderly patients with delirium, functional decline or diarrhoea. This likely explains the greater number of staff infected in several acute medical wards (shown in green near the top of *Figure 8B*), compared to Covid-19 cohort wards (shown in red).

The reported rates of exposure without PPE were similar among medical and ICU staff (42% and 38%, *Supplementary file 1E*), likely reflecting exposures to ICU staff visiting wards to assess critically ill patients. Universal admission testing was only introduced on 24th April 2020, and the limited availability and speed of testing in the early phase of the pandemic likely delayed identification of some Covid-19 cases.

It is difficult to say whether level-1 PPE was less protective than level-2. Increased Covid-19 in staff reporting exposure to a Covid-19 patient without PPE suggests surgical masks afford some

**Table 2.** Local recommendations for PPE and testing, based on contemporaneous national Public Health England guidance.

| | PPE | Testing |
|---|---|---|
| Until 25th February 2020 | Full length gown, gloves, correctly fitted FFP3 mask and full face visor (level-2 PPE)<br>Side room isolation | Clinical syndrome with relevant travel history |
| 25th February | As above for suspected cases with travel history<br>For severe community acquired pneumonia without travel history: gown/apron, gloves, and fluid repellent mask (FFP3 for aerosol generating procedures); no need to isolate pending test | Clinical syndrome with relevant travel history<br>Severe community acquired pneumonia |
| 8th March | Fluid resistant surgical masks, gloves, apron and optional eye protection for symptomatic but unconfirmed inpatients (level-1 PPE). Eye protection to be worn if risk of eye contamination.<br>Full level-2 PPE for confirmed cases and Aerosol Generating Procedures (AGPs). | Clinical syndrome with relevant travel history<br>Severe community acquired pneumonia |
| 13th March | Fluid resistant surgical masks, gloves, apron and risk assessment for eye protection for suspected and confirmed Covid-19 inpatients (level-1 PPE)<br>Surgical masks on entry to Covid-19 cohort wards<br>Apron, gloves and FFP3 mask on intensive care<br>FFP3 mask, disposable gown, eye protection and gloves for AGPs | Any respiratory illness requiring admission to hospital and either radiological evidence of pneumonia or ARDS or influenza-like illness with fever > 37.8C |
| 14-16th March | As above<br>All suspected Covid-19 patients admitted directly via acute medicine (bypassing the emergency department) | Any influenza like illness |
| 1st April | Universal minimum level-1 PPE across all wards<br>Level-2 PPE for AGPs as above | |
| 6th April | AGPs: gloves, disposable gown, FFP3 mask, eye protection<br>Working in higher risk area (ICU/High Dependency Unit) with confirmed cases: gloves, apron, gown, FFP3 mask and eye protection<br>Level-1 PPE elsewhere | Diagnosis based on either positive swab or 'Covid-19 syndrome' (influenza like illness and compatible radiology and no alternative explanation) |
| 24th April | | Universal admission testing for all patients irrespective of clinical syndrome |

protection, and protection from influenza has been reported to be similar using surgical versus FFP2 masks.(*Radonovich et al., 2019*) However, it is likely that a bundle of measures (level-2 PPE, training, supervision and space for donning and doffing, increased staffing levels) influenced the lower risk in ICU and HDU staff (*Figure 3*, *Figure 8*). As with many infection control intervention-bundles, it is difficult to distinguish which component was most important.

It is also likely that staff-to-staff transmission amplified incidence, based on high Covid-19 rates in several wards without large numbers of Covid-19 patients. Future viral genome sequencing studies may allow analysis of the relative contribution of patients and staff to transmission.

Increased risk of adverse outcomes has been widely reported in Black and Asian ethnic groups (*Williamson et al., 2020*), with evidence they are also at increased risk of infection (*Disparities in the risk and outcomes from COVID-19, 2020*; *Khunti et al., 2020*). Here, we show Black and Asian staff were at greater risk of infection after controlling for age, gender, working location, role, and exposure at home. Job role can be thought of as a proxy for socio-economic background but we were not able to control directly for income levels, home circumstances, pre-morbid conditions or other potential structural inequalities. That staff working as porters or cleaners had the greatest adjusted risk of infection is consistent with economics playing a part in risk, potentially reflecting conditions outside of the hospital, for example dense occupancy of living space due to lower incomes.

Multiple complex causal relationships are included within the multivariable model. For example, ethnicity via structural inequalities may influence occupational role, speciality and exposures outside the hospital, which all subsequently influence infection risk. Within the multivariable model the aOR for ethnicity only represents the part of the infection risk associated with ethnicity that is not mediated by the other factors in the model. As such, the overall impact of ethnicity in the context of current structural inequalities may be better captured by the univariable OR. Similarly, the aOR reported for speciality represents the speciality specific risk that is not mediated via the Covid-facing status of the healthcare workers which is included separately. As most specialty-specific risk in our

model was mediated by working in a Covid-19 area rather than by the risks of the speciality per se (e.g. proximity to patient airways), specialities working in Covid-19 areas may appear less at risk if the aORs for each speciality are considered in isolation. Instead, to calculate a personalised Covid-19 risk score, all factors in the multivariable model need to be considered together, that is simultaneously adjusting for all the relevant separate aOR present. For example, an Asian Covid-19-facing medical nurse is 7.64 (95%CI 5.60–10.43) more likely to be infected than a white non-Covid-19-facing administrative worker. Notably, this exceeds the risk of living with someone with known Covid-19 (aOR 4.82, 95% CI 3.45–6.72).

We observed 24% of staff/patients remained PCR-positive at ≥6 weeks post-diagnosis. Fewer staff than patients were persistently positive at 7–13 days, potentially reflecting greater time from infection to initial diagnosis in asymptomatic staff compared to symptomatic patients, and/or milder infection in staff. However, the proportions of patients and staff persistently positive were similar from ≥14 days onwards.

Limitations of our study include its cross-sectional nature, with most staff diagnosed retrospectively using serology. As a result, we can only partially reconstruct week-by-week incidence in staff on the basis of contemporaneous testing of symptomatic staff and retrospective staff recall of symptom onset in asymptomatic staff diagnosed once recovered who were unwell before symptomatic staff testing was widely available (*Figure 1B*). The lack information on when most staff were infected also makes it challenging to reconstruct the source of individual staff infections, for example other staff or patients, and to analyze how this varied over time with PPE changes and other infection control interventions.

It is also unknown what proportion of staff who were infected either mounted no detectable antibody response or in whom it had waned by the time of testing. Despite the cross-sectional design, the numbers of staff tested, meant that testing spanned a seven week period from late April onwards, potentially leading to confounding by week of testing with changes in incidence over time and possible variation in staff groups attending testing. However, we did not see a change in our composite end point of SARS-CoV-2 PCR and/or IgG positive by week of testing, likely because asymptomatic staff testing was undertaken after the peak in incidence at the end of March and beginning of April 2020.

As our testing programme was voluntary, it is possible that different staff groups participated at different rates. For example, if staff differentially attended based on whether they believed they had already had Covid-19 this may have led to selection bias. However, rates of staff participation were high overall, with 77% registering to participate and 73% attending for a test.

Additionally, the data gathered on particular exposures may be subject to recall bias. Several risk factors were invariant over the time of the study including gender, ethnicity, approximate age and for most staff their role, specialty and working location. However, exposure histories such as living with someone with suspected Covid-19 or Covid-19 exposure without PPE at work maybe subject to recall bias. To mitigate this, when considering workplace exposure without PPE we present sensitivity analyses in the subset of staff who did not believe they had previously had Covid-19. Our data are also from a single setting and findings may vary by practice, geography and population-wide Covid-19 incidence (*Shields et al., 2020*; *Houlihan et al., 2020*).

Our study suggests that an earlier move to universal level-1 PPE may have prevented some infections and that a consistent bundle of level-2 PPE provision and use, training, and supervision and space for donning and doffing protected staff working in high-risk areas. Wider deployment of this bundle should be considered where staff are at increased risk. Our study provides data to inform risk assessments for staff, to ensure those staff most at risk are deployed appropriately. Given likely staff-to-staff transmission where COVID-19 patient pressure was low, there is a need to protect all staff regardless of role. This includes reinforcement of measures to support social distancing and raises questions about the role of social inequality in Covid-19 transmission. If some staff are already immune the impact of any future Covid-19 surge may be less marked for staff, although differential deployment or use of PPE based on immune status would require evidence it was safe and socially acceptable. Our testing programme has been highly popular with staff, ensured enhanced detection of those with Covid-19, and now also provides a large cohort to inform studies on the extent of antibody-mediated protection against future infection.

# Materials and methods

## Setting and data collection

OUH spans four teaching hospitals with 1000 beds and 13,800 staff, serving a population of 680,000 and acting as a regional referral centre. The first patients with Covid-19 were admitted to OUH in mid-March 2020. SARS-CoV-2 testing, initially reserved for inpatients, was extended to symptomatic staff and staff household contacts with fever (≥37.8°C) or new-onset cough from 27th March. Testing for symptomatic staff and symptomatic staff household contacts was offered by the hospital's Occupational Health department between days 2 and 4 of symptoms, only PCR results from staff are presented. Staff awaiting a test or test result were asked to self-isolate at home. From 18th May 2020 onwards, testing criteria were expanded to include staff with new onset anosmia. In line with national guidance, staff without these specific symptoms (fever, cough, anosmia) were considered unlikely to have Covid-19 and permitted to remain at work.

A voluntary asymptomatic screening programme for all staff working anywhere on site commenced on 23rd April. All staff not meeting the criteria for symptomatic testing were considered eligible for asymptomatic testing. Both naso- and oro-pharyngeal swabs were obtained from each staff member for real-time-PCR for SARS-CoV-2 and blood for serological analysis by specially-trained nurses, medical students and other healthcare professionals. Appointments were available up to six days a week across all hospitals, with staff required to register details on a bespoke website within the NHS network prior to booking. Data were collected on age, self-reported gender and ethnicity, role, working location and history of symptoms, whether they were patient facing, and whether they had at any time been exposed to a patient with Covid-19 without any PPE. Staff were asked whether they believed they had had Covid-19 already, and whether they had had household or community-based contact with a suspected or confirmed Covid-19 case.

Automated reporting of results was followed-up with a phone call for positive PCR results to distinguish contemporaneous from previous infection (>7 days ago). The former were asked to self-isolate for seven days, and their household contacts for 14 days.

## Infection control

From 1st February 2020, 'level-2 PPE' (full length gown, gloves, correctly fitted FFP3/N99 mask and full face visor) was mandated for any contact with a confirmed or suspected case. From 8th March this was downgraded to 'level-1 PPE' (fluid resistant surgical mask, gloves, apron and optional eye protection), except for aerosol generating procedures.(*Aerosol Generating Procedures, 2020*) From 1st April a minimum of level-1 PPE was mandated for all patient care, regardless of Covid-19 status (*Table 2*).

## Laboratory assays

RT-PCR was performed at OUH using the PHE SARS-CoV-2 assay (targeting the RdRp gene), or one of two commercial assays: Abbott RealTime (targeting RdRp and N genes; Abbott, Maidenhead, UK), Altona RealStar (targeting E and S genes; Altona Diagnostics, Liverpool, UK), or using the ABI 7500 platform (Thermo Fisher, Abingdon, UK) with the US Centers for Disease Control and Prevention Diagnostic Panel of two probes targeting the N gene. Samples from 2 days of testing were processed by the UK Lighthouse Labs network (Milton Keynes) using the Thermo Fisher TaqPath assay (targeting S and N genes, and ORF1ab; Thermo Fisher, Abingdon, UK).

Serological investigations were performed by chemiluminescent microparticle immunoassay (CMIA) for IgG to nucleocapsid protein on Abbott Architect (Abbott, Maidenhead, UK) with a manufacturer's signal-to-cut-off index of 1.4, and an enzyme-linked immunosorbent assay (ELISA) platform developed at the Target Discovery Institute (University of Oxford) detecting IgG to trimeric spike antigen, using net-normalised signal cut-off of 8 million (*The National SARS-CoV-2 Serology Assay Evaluation Group, 2020*; *Adams et al., 2020*).

## Statistical analysis

Univariable and multivariable logistic regression was performed to assess risk factors for infection using a composite endpoint of 'Covid-19 at any time', based on a positive RT-PCR test or the detection of IgG by ELISA and/or CMIA. Natural cubic splines were used to account for non-linear

relationships with continuous variables. Given the number of potential predictors fitted, backwards model selection was undertaken using AIC values. We screened for first-order interactions between main effects using a Wald p-value threshold of <0.01. We analysed risk factors for detection of SARS-CoV-2 IgG antibodies using the same approach.

Similarly, univariable and multivariable logistic regression, was used to assess associations between 'Covid-19 at any time' and 11 self-reported symptoms prior to testing. As only 11 potential predictors were included in the model variable selection was not undertaken.

Univariable and multivariable linear regression was used to assess the relationship between ward-based Covid-19 patient infectious pressure and the proportion of staff working on a ward with Covid-19. Covid-19 infectious pressure was calculated by considering each patient infectious from −2 to +7 days around the date of their first positive SARS-CoV-2 PCR test. Only staff working in a single ward were included in the analysis.

Analyses were performed using R, version 3.6.3.

## Acknowledgements

We are extremely grateful to all the NHS staff who participated in our programme and provided data and samples. We would like to pay tribute to all the staff working at the Oxford University Hospitals NHS Foundation Trust, and their families, and in particular to those who became seriously unwell and the four staff members who died from Covid-19. This work uses data provided by patients and staff and collected by the UK's National Health Service as part of their care and support. We thank all the people of Oxfordshire who contribute to the Infections in Oxfordshire Research Database. Research Database Team: L Butcher, H Boseley, C Crichton, DW Crook, DW Eyre, O Freeman, J Gearing (community), R Harrington, K Jeffery, M Landray, A Pal, TEA Peto, TP Quan, J Robinson (community), J Sellors, B Shine, AS Walker, D Waller. Patient and Public Panel: G Blower, C Mancey, P McLoughlin, B Nichols.

**Oxford University Hospitals Staff Testing Group**:

University of Oxford Medical School staff testing team (University of Oxford, Oxford, UK): Adam J R Watson, Adan Taylor, Alan Chetwynd, Alexander Grassam-Rowe, Alexandra S Mighiu, Angus Livingstone, Annabel Killen, Caitlin Rigler, Callum Harries, Cameron East, Charlotte Lee, Chris J B Mason, Christian Holland, Connor Thompson, Conor Hennesey, Constantinos Savva, David S Kim, Edward W A Harris, Euan J McGivern, Evelyn Qian, Evie Rothwell, Francesca Back, Gabriella Kelly, Gareth Watson, Gregory Howgego, Hannah Chase, Hannah Danbury, Hannah Laurenson-Schafer, Harry L Ward, Holly Hendron, Imogen C Vorley, Isabel Tol, James Gunnell, Jocelyn LF Ward, Jonathan Drake, Joseph D Wilson, Joshua Morton, Julie Dequaire, Katherine O'Byrne, Kenzo Motohashi, Kirsty Harper, Krupa Ravi, Lancelot J Millar, Liam J Peck, Madeleine Oliver, Marcus Rex English, Mary Kumarendran, Matthew Wedlich, Olivia Ambler, Oscar T Deal, Owen Sweeney, Philip Cowie, Rebecca te Water Naudé, Rebecca Young, Rosie Freer, Samuel Scott, Samuel Sussmes, Sarah Peters, Saxon Pattenden, Seren Waite, Síle Ann Johnson, Stefan Kourdov, Stephanie Santos-Paulo, Stoyan Dimitrov, Sven Kerneis, Tariq Ahmed-Firani, Thomas B King, Thomas G Ritter, Thomas H Foord, Zoe De Toledo, Thomas Christie

Oxford University Hospitals staff testing team (Oxford University Hospitals NHS Foundation Trust, Oxford, UK): Bernadett Gergely, David Axten, Emma-Jane Simons, Heather Nevard, Jane Philips, Justyna Szczurkowska, Kaisha Patel, Kyla Smit, Laura Warren, Lisa Morgan, Lucianne Smith, Maria Robles, Mary McKnight, Michael Luciw, Michelle Gates, Nellia Sande, Rachel Turford, Roshni Ray, Sonam Rughani, Tracey Mitchell, Trisha Bellinger, Vicki Wharton

Oxford University Hospitals microbiology laboratory (Oxford University Hospitals NHS Foundation Trust, Oxford, UK): Anita Justice, Gerald Jesuthasan, Susan Wareing, Nurul Huda Mohamad Fadzillah, Kathryn Cann, Richard Kirton

Oxford University Hospitals Infection, Prevention and Control team (Oxford University Hospitals NHS Foundation Trust, Oxford, UK): Claire Sutton, Claudia Salvagno, Gabriella D'Amato, Gemma Pill, Lisa Butcher, Lydia Rylance-Knight, Merline Tabirao, Ruth Moroney, Sarah Wright.

# Additional information

## Group author details

**Oxford University Hospitals Staff Testing Group**

**Adam JR Watson**: University of Oxford, Oxford, United Kingdom; **Adan Taylor**: University of Oxford, Oxford, United Kingdom; **Alan Chetwynd**: University of Oxford, Oxford, United Kingdom; **Alexander Grassam-Rowe**: University of Oxford, Oxford, United Kingdom; **Alexandra S Mighiu**: University of Oxford, Oxford, United Kingdom; **Angus Livingstone**: University of Oxford, Oxford, United Kingdom; **Annabel Killen**: University of Oxford, Oxford, United Kingdom; **Caitlin Rigler**: University of Oxford, Oxford, United Kingdom; **Callum Harries**: University of Oxford, Oxford, United Kingdom; **Cameron East**: University of Oxford, Oxford, United Kingdom; **Charlotte Lee**: University of Oxford, Oxford, United Kingdom; **Chris JB Mason**: University of Oxford, Oxford, United Kingdom; **Christian Holland**: University of Oxford, Oxford, United Kingdom; **Connor Thompson**: University of Oxford, Oxford, United Kingdom; **Conor Hennesey**: University of Oxford, Oxford, United Kingdom; **Constantinos Savva**: University of Oxford, Oxford, United Kingdom; **David S Kim**: University of Oxford, Oxford, United Kingdom; **Edward WA Harris**: University of Oxford, Oxford, United Kingdom; **Euan J McGivern**: University of Oxford, Oxford, United Kingdom; **Evelyn Qian**: University of Oxford, Oxford, United Kingdom; **Evie Rothwell**: University of Oxford, Oxford, United Kingdom; **Francesca Back**: University of Oxford, Oxford, United Kingdom; **Gabriella Kelly**: University of Oxford, Oxford, United Kingdom; **Gareth Watson**: University of Oxford, Oxford, United Kingdom; **Gregory Howgego**: University of Oxford, Oxford, United Kingdom; **Hannah Chase**: University of Oxford, Oxford, United Kingdom; **Hannah Danbury**: University of Oxford, Oxford, United Kingdom; **Hannah Laurenson-Schafer**: University of Oxford, Oxford, United Kingdom; **Harry L Ward**: University of Oxford, Oxford, United Kingdom; **Holly Hendron**: University of Oxford, Oxford, United Kingdom; **Imogen C Vorley**: University of Oxford, Oxford, United Kingdom; **Isabel Tol**: University of Oxford, Oxford, United Kingdom; **James Gunnell**: University of Oxford, Oxford, United Kingdom; **Jocelyn LF Ward**: University of Oxford, Oxford, United Kingdom; **Jonathan Drake**: University of Oxford, Oxford, United Kingdom; **Joseph D Wilson**: University of Oxford, Oxford, United Kingdom; **Joshua Morton**: University of Oxford, Oxford, United Kingdom; **Julie Dequaire**: University of Oxford, Oxford, United Kingdom; **Katherine O'Byrne**: University of Oxford, Oxford, United Kingdom; **Kenzo Motohashi**: University of Oxford, Oxford, United Kingdom; **Kirsty Harper**: University of Oxford, Oxford, United Kingdom; **Krupa Ravi**: University of Oxford, Oxford, United Kingdom; **Lancelot J Millar**: University of Oxford, Oxford, United Kingdom; **Liam J Peck**: University of Oxford, Oxford, United Kingdom; **Madeleine Oliver**: University of Oxford, Oxford, United Kingdom; **Marcus Rex English**: University of Oxford, Oxford, United Kingdom; **Mary Kumarendran**: University of Oxford, Oxford, United Kingdom; **Matthew Wedlich**: University of Oxford, Oxford, United Kingdom; **Olivia Ambler**: University of Oxford, Oxford, United Kingdom; **Oscar T Deal**: University of Oxford, Oxford, United Kingdom; **Owen Sweeney**: University of Oxford, Oxford, United Kingdom; **Philip Cowie**: University of Oxford, Oxford, United Kingdom; **Rebecca te Water Naudé**: University of Oxford, Oxford, United Kingdom; **Rebecca Young**: University of Oxford, Oxford, United Kingdom; **Rosie Freer**: University of Oxford, Oxford, United Kingdom; **Samuel Scott**: University of Oxford, Oxford, United Kingdom; **Samuel Sussmes**: University of Oxford, Oxford, United Kingdom; **Sarah Peters**: University of Oxford, Oxford, United Kingdom; **Saxon Pattenden**: University of Oxford, Oxford, United Kingdom; **Seren Waite**: University of Oxford, Oxford, United Kingdom; **Síle Ann Johnson**: University of Oxford, Oxford, United Kingdom; **Stefan Kourdov**: University of Oxford, Oxford, United Kingdom; **Stephanie Santos-Paulo**: University of Oxford, Oxford, United Kingdom; **Stoyan Dimitrov**: University of Oxford, Oxford, United Kingdom; **Sven Kerneis**: University of Oxford, Oxford, United Kingdom; **Tariq Ahmed-Firani**: University of Oxford, Oxford, United Kingdom; **Thomas B King**: University of Oxford, Oxford, United Kingdom; **Thomas G Ritter**: University of Oxford, Oxford, United Kingdom; **Thomas H Foord**: University of Oxford, Oxford, United Kingdom; **Zoe De Toledo**: University of Oxford, Oxford, United Kingdom; **Thomas Christie**: University of Oxford, Oxford, United Kingdom; **Bernadett Gergely**: Oxford University Hospitals NHS Foundation Trust, Oxford, United Kingdom; **David Axten**: Oxford University Hospitals NHS Foundation Trust, Oxford, United Kingdom; **Emma-Jane Simons**: Oxford University Hospitals NHS Foundation Trust, Oxford, United Kingdom; **Heather**

**Nevard**: Oxford University Hospitals NHS Foundation Trust, Oxford, United Kingdom; **Jane Philips**: Oxford University Hospitals NHS Foundation Trust, Oxford, United Kingdom; **Justyna Szczurkowska**: Oxford University Hospitals NHS Foundation Trust, Oxford, United Kingdom; **Kaisha Patel**: Oxford University Hospitals NHS Foundation Trust, Oxford, United Kingdom; **Kyla Smit**: Oxford University Hospitals NHS Foundation Trust, Oxford, United Kingdom; **Laura Warren**: Oxford University Hospitals NHS Foundation Trust, Oxford, United Kingdom; **Lisa Morgan**: Oxford University Hospitals NHS Foundation Trust, Oxford, United Kingdom; **Lucianne Smith**: Oxford University Hospitals NHS Foundation Trust, Oxford, United Kingdom; **Maria Robles**: Oxford University Hospitals NHS Foundation Trust, Oxford, United Kingdom; **Mary McKnight**: Oxford University Hospitals NHS Foundation Trust, Oxford, United Kingdom; **Michael Luciw**: Oxford University Hospitals NHS Foundation Trust, Oxford, United Kingdom; **Michelle Gates**: Oxford University Hospitals NHS Foundation Trust, Oxford, United Kingdom; **Nellia Sande**: Oxford University Hospitals NHS Foundation Trust, Oxford, United Kingdom; **Rachel Turford**: Oxford University Hospitals NHS Foundation Trust, Oxford, United Kingdom; **Roshni Ray**: Oxford University Hospitals NHS Foundation Trust, Oxford, United Kingdom; **Sonam Rughani**: Oxford University Hospitals NHS Foundation Trust, Oxford, United Kingdom; **Tracey Mitchell**: Oxford University Hospitals NHS Foundation Trust, Oxford, United Kingdom; **Trisha Bellinger**: Oxford University Hospitals NHS Foundation Trust, Oxford, United Kingdom; **Vicki Wharton**: Oxford University Hospitals NHS Foundation Trust, Oxford, United Kingdom; **Anita Justice**: Oxford University Hospitals NHS Foundation Trust, Oxford, United Kingdom; **Gerald Jesuthasan**: Oxford University Hospitals NHS Foundation Trust, Oxford, United Kingdom; **Susan Wareing**: Oxford University Hospitals NHS Foundation Trust, Oxford, United Kingdom; **Nurul Huda Mohamad Fadzillah**: Oxford University Hospitals NHS Foundation Trust, Oxford, United Kingdom; **Kathryn Cann**: Oxford University Hospitals NHS Foundation Trust, Oxford, United Kingdom; **Richard Kirton**: Oxford University Hospitals NHS Foundation Trust, Oxford, United Kingdom; **Claire Sutton**: Oxford University Hospitals NHS Foundation Trust, Oxford, United Kingdom; **Claudia Salvagno**: Oxford University Hospitals NHS Foundation Trust, Oxford, United Kingdom; **Gabriella DAmato**: Oxford University Hospitals NHS Foundation Trust, Oxford, United Kingdom; **Gemma Pill**: Oxford University Hospitals NHS Foundation Trust, Oxford, United Kingdom; **Lisa Butcher**: Oxford University Hospitals NHS Foundation Trust, Oxford, United Kingdom; **Lydia Rylance-Knight**: Oxford University Hospitals NHS Foundation Trust, Oxford, United Kingdom; **Merline Tabirao**: Oxford University Hospitals NHS Foundation Trust, Oxford, United Kingdom; **Ruth Moroney**: Oxford University Hospitals NHS Foundation Trust, Oxford, United Kingdom; **Sarah Wright**: Oxford University Hospitals NHS Foundation Trust, Oxford, United Kingdom

## Competing interests

David W Eyre: Lecture fees from Gilead, outside the submitted work. The other authors declare that no competing interests exist.

## Funding

| Funder | Grant reference number | Author |
| --- | --- | --- |
| UK Government | Department of Health and Social Care | David W Eyre<br>Sheila F Lumley<br>Denise O'Donnell<br>Mark Campbell<br>Elizabeth Sims<br>Elaine Lawson<br>Fiona Warren<br>Tim James<br>Stuart Cox<br>Alison Howarth<br>George Doherty<br>Stephanie B Hatch<br>James Kavanagh<br>Kevin K Chau<br>Philip W Fowler<br>Jeremy Swann<br>Denis Volk<br>Fan Yang-Turner<br>Nicole Stoesser<br>Philippa C Matthews<br>Maria Dudareva<br>Timothy Davies<br>Robert H Shaw<br>Leon Peto<br>Louise O Downs<br>Alexander Vogt<br>Ali Amini<br>Bernadette C Young<br>Philip George Drennan<br>Alexander J Mentzer<br>Donal T Skelly<br>Fredrik Karpe<br>Matt J Neville<br>Monique Andersson<br>Andrew J Brent<br>Nicola Jones<br>Lucas Martins Ferreira<br>Thomas Christott<br>Brian D Marsden<br>Sarah Hoosdally<br>Richard Cornall<br>Derrick W Crook<br>David I Stuart<br>Gavin Screaton<br>Timothy EA Peto<br>Bruno Holthof<br>Anne-Marie O'Donnell<br>Daniel Ebner |
| National Institute of Health Research Health Protection Research Unit in Healthcare Associated Infections and Antimicrobial Resistance | HPRU-2012-10041 | David W Eyre<br>Sheila F Lumley<br>Denise O'Donnell<br>Mark Campbell<br>Elizabeth Sims<br>Elaine Lawson<br>Fiona Warren<br>Tim James<br>Stuart Cox<br>Alison Howarth<br>George Doherty |
| Robertson Foundation | | David W Eyre |
| NIHR | Oxford BRC Senior Fellow | David W Eyre<br>Philippa C Matthews |
| Wellcome Trust | Clinical Research Fellow | Sheila F Lumley |
| Medical Research Council | MR/N00065X/1 | David I Stuart |
| Wellcome Trust | Intermediate Fellowship 110110/Z/15/Z | Philippa C Matthews |

| NIHR | Doctoral Research Fellow | Maria Dudareva |
| --- | --- | --- |
| Medical Research Foundation | MRF-145-004-TPG-AVISO | Kevin K Chau |
| Wellcome Trust | Clinical Research Training Fellow 216417/Z/19/Z | Ali Amini |
| NIHR | Clinical Lecturer | Bernadette C Young |
| Structural Genomics Consortium | | Lucas Martins Ferreira Thomas Christott Brian D Marsden |
| Kennedy Trust for Rheumatology Research | | Brian D Marsden |
| Wellcome Trust | Senior Investigator | Gavin Screaton |
| Schmidt Foundation | | Gavin Screaton |
| Wellcome Trust Career Development Fellow | 214560/Z/18/Z | Timothy M Walker |

The funders had no role in study design, data collection and interpretation, or the decision to submit the work for publication.

## Author contributions

David W Eyre, Conceptualization, Data curation, Software, Formal analysis, Supervision, Investigation, Visualization, Writing - original draft, Project administration, Writing - review and editing; Sheila F Lumley, Denise O'Donnell, Tim James, Alison Howarth, Investigation, Project administration; Mark Campbell, Formal analysis, Investigation; Elizabeth Sims, Elaine Lawson, Fiona Warren, Stuart Cox, George Doherty, Stephanie B Hatch, James Kavanagh, Kevin K Chau, Maria Dudareva, Timothy Davies, Robert H Shaw, Leon Peto, Louise O Downs, Alexander Vogt, Ali Amini, Bernadette C Young, Philip George Drennan, Alexander J Mentzer, Donal T Skelly, Fredrik Karpe, Matt J Neville, Oxford University Hospitals Staff Testing Group, Investigation; Philip W Fowler, Jeremy Swann, Denis Volk, Fan Yang-Turner, Lucas Martins Ferreira, Thomas Christott, Software; Nicole Stoesser, Philippa C Matthews, Investigation, Methodology; Monique Andersson, Richard Cornall, Derrick W Crook, Anne-Marie O'Donnell, Supervision; Andrew J Brent, Nicola Jones, Christopher P Conlon, Conceptualization, Supervision; Brian D Marsden, Software, Project administration; Sarah Hoosdally, Project administration; David I Stuart, Gavin Screaton, Resources, Supervision, Methodology; Timothy EA Peto, Conceptualization, Data curation, Formal analysis, Supervision, Investigation; Bruno Holthof, Funding acquisition; Daniel Ebner, Resources, Investigation, Methodology, Project administration; Katie Jeffery, Conceptualization, Formal analysis, Supervision, Funding acquisition, Writing - review and editing; Timothy M Walker, Conceptualization, Data curation, Formal analysis, Supervision, Funding acquisition, Investigation, Writing - original draft, Project administration, Writing - review and editing

## Author ORCIDs

David W Eyre  https://orcid.org/0000-0001-5095-6367
Philip W Fowler  https://orcid.org/0000-0003-0912-4483
Nicole Stoesser  http://orcid.org/0000-0002-4508-7969
Bernadette C Young  http://orcid.org/0000-0001-6071-6770
Philip George Drennan  http://orcid.org/0000-0003-3367-3335
Alexander J Mentzer  https://orcid.org/0000-0002-4502-2209
Donal T Skelly  https://orcid.org/0000-0002-2426-3097
Matt J Neville  http://orcid.org/0000-0002-6004-5433
Brian D Marsden  http://orcid.org/0000-0002-1937-4091
Derrick W Crook  http://orcid.org/0000-0002-0590-2850
Daniel Ebner  http://orcid.org/0000-0002-6495-7026

### Ethics

Human subjects: All asymptomatic staff data collection and testing were part of enhanced hospital infection prevention and control measures instituted by the UK Department of Health and Social Care (DHSC). Deidentified data from staff testing and patients were obtained from the Infections in Oxfordshire Research Database (IORD) which has generic Research Ethics Committee, Health Research Authority and Confidentiality Advisory Group approvals (19/SC/0403, ECC5-017(A)/2009). De-identified patient data extracted included admission and discharge dates, ward location and positive Covid-19 test results.

### Decision letter and Author response

Decision letter https://doi.org/10.7554/eLife.60675.sa1
Author response https://doi.org/10.7554/eLife.60675.sa2

## Additional files

### Supplementary files

• Supplementary file 1. Tables. (A) Comparison of serology results by two methods, the Abbott Architect i2000 and Target Discovery Institute ELISA. (B) Univariable and multivariable relationships between risk factors and staff infection with SARS-CoV-2 in 10,032 healthcare workers. (C) Multivariable relationships between risk factors and SARS-CoV-2 IgG positivity in 9956 healthcare workers. (D) Multivariable linear regression, relationship between the percentage of staff with Covid-19 and ward-based Covid-19 pressure and ward type. (E) Rates of self-reported exposure without PPE by staff specialty.

• Transparent reporting form

### Data availability

The data studied are available from the Infections in Oxfordshire Research Database (https://oxfordbrc.nihr.ac.uk/research-themes-overview/antimicrobial-resistance-and-modernising-microbiology/infections-in-oxfordshire-research-database-iord/), subject to an application and research proposal meeting the ethical and governance requirements of the Database. For further details on how to apply for access to the data and for a research proposal template please email iord@ndm.ox.ac.uk.

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
