## [Decision Letter]

**Acceptance summary:**

This is a comprehensive and careful study of risk factors for COVID-19 infections in HCW in Oxford hospitals. While the design reflects some of the exigencies of data gathering in an emergency, it is carefully analyzed and the caveats are clearly specified.

**Decision letter after peer review:**

Thank you for submitting your article "Differential occupational risks to healthcare workers from SARS-CoV-2 observed during a prospective observational study" for consideration by *eLife*. Your article has been reviewed by three peer reviewers, one of whom is a member of our Board of Reviewing Editors, and the evaluation has been overseen by Miles Davenport as the Senior Editor. The following individual involved in review of your submission has agreed to reveal their identity: M Estee Torok (Reviewer #3).

The reviewers have discussed the reviews with one another and the Reviewing Editor has drafted this decision to help you prepare a revised submission.

Summary:

This was a prospective observational cohort study COVID-19 testing of over 9,000 staff at a UK teaching hospital. The investigators tested symptomatic and asymptomatic staff using nasopharyngeal PCR and immunoassays for SARS-CoV-2. 11% of staff were found to have evidence of current or previous SARS-CoV-2 infection. Analysis of risk factors identified household contact with a confirmed positive case to be the greatest risk. Higher rates were also seen in staff working in COVID-facing areas, particularly in acute medicine areas, and in certain ethnic groups (Black and Asian) and occupational groups (e.g. porters and cleaners). Lower rates were seen in ICU staff, likely reflecting PPE measures.

Major comments:

1) The major issue with this analysis is that there is inadequate control for, and even discussion of, three sorts of potential biases: confounding, selection bias, and measurement error. These are intertwined in complex ways because of the endpoint of the analyses which is composite in three different ways: cumulative (serologic) vs. current (PCR) infection, multiple time points (even putting aside the fact that some were tested more than once), and many different reasons for testing. This creates potential confounding because, all else equal, the probability of infection is highest in the middle of the study for PCR and at the end for serology; at many points it is higher for serology than for PCR. If certain groups of employees were more likely to get serology tests at the end, this would elevate their rates of the outcome. It could create selection bias because the denominator is those tested, not total employees in a category (I think, not completely clear), and different groups may have opted for testing at different rates. It creates measurement error (closely related to the confounding) because the test types have different sensitivities and specificities / NPV and PPV.

An ideal analysis would adjust for time (by perhaps week) and test type and the interaction (since the shape of the time effect is quite different by test type). There would also be clearer discussion and perhaps adjustment for the issue below in point 2. This may be impossible even with this large data set, but some kind of effort needs to be made. Otherwise the risk factor findings, plausible as they are, are not very robust. I would personally be convinced if these more nuanced analyses showed the same patterns as the overall analysis, even if not as statistically convincing. An overhaul of the analysis of risk factors is needed.

2) The other analyses, as well as the risk factor analyses, are confusing in that it is very hard to follow who is being tested with no history of symptoms, past symptoms, future symptoms, etc. Especially confusing is Figure 1 which seems to have only two categories – symptomatic or previously symptomatic, even after asymptomatic testing starts. As stated by another reviewer:

In addition, I had a number of questions about what governed the testing and how the distribution of cases changed over time.

– For the duration of PCR positivity, what determined whether individuals had repeat tests?

– Given the variation in infection control and PPE efforts, how were presumed workplace acquisitions distributed across time? And same with staff roles?

– Were the symptoms that enabled staff to get tested restricted to fever and cough for the duration of the study period?

– For the asymptomatic screening program, did "asymptomatic" mean no fever and cough, or none of the other symptoms associated with COVID-19?

– For PCR testing, were either NP or OP swabs obtained, or both?

– How incomplete were each of the fields in the reports from participants in the asymptomatic screening program? It seems only 52% reported a date of symptoms. Were all the other fields completed by all participants?

– It seems a number of assays were used. How variable were their test characteristics? Was there any variation in their use over time?

3) Overall, it is also really unclear what outcome is being studied where, exacerbated by the inconsistent headings vs. text (e.g. serologic results heading followed by results on PCR or serologic outcomes).

4) Another limitation of the study is that it was cross-sectional in nature and data on particular exposures may have been subject to recall bias. A potential solution to this would have been to conduct patient and staff screening in parallel from the start of the epidemic but this was not feasible. This should be discussed, at least.

[Editors' note: further revisions were suggested prior to acceptance, as described below.]

Thank you for submitting your article "Differential occupational risks to healthcare workers from SARS-CoV-2 observed during a prospective observational study" for consideration by *eLife*. Your article has been reviewed and the evaluation has been overseen by a Reviewing Editor and Miles Davenport as the Senior Editor.

The Reviewing Editor has drafted this decision to help you prepare a revised submission to address some small issues.

We would like to draw your attention to changes in our revision policy that we have made in response to COVID-19 (https://elifesciences.org/articles/57162). Specifically, we are asking editors to accept without delay manuscripts, like yours, that they judge can stand as *eLife* papers without additional data, even if they feel that they would make the manuscript stronger. Thus, the revisions requested below only address clarity and presentation.

The revision responds adequately to the issues raised, and we have discussed two small changes that would make the paper ready for publication:

1) Put the error-bar figures (now supplements) in the main text.

2) Talk about the multivariable model and its meanings. Most of the "adjustments" are in fact blocking causal pathways: for example, ethnicity -> role -> infection risk in the multivariable model is blocked by including role, making the multivariable model less indicative in some cases of likely causal relationships than the univariable ones. Likewise, specialty-> patient-facing (or exposure) -> infection is more meaningfully capturing the specialty effect in univariate (Not conditioning on patient-facing or exposure) than multi. Since many people think multivariable are the "real" effect measures this should be mentioned. The Discussion should convey what the data imply about causality, assuming that unmeasured confounding and other sources of bias are adequately mitigated, this will not be obvious without some explication.

---

## [Author Response]

Major comments:1) The major issue with this analysis is that there is inadequate control for, and even discussion of, three sorts of potential biases: confounding, selection bias, and measurement error. These are intertwined in complex ways because of the endpoint of the analyses which is composite in three different ways: cumulative (serologic) vs. current (PCR) infection, multiple time points (even putting aside the fact that some were tested more than once), and many different reasons for testing. This creates potential confounding because, all else equal, the probability of infection is highest in the middle of the study for PCR and at the end for serology; at many points it is higher for serology than for PCR. If certain groups of employees were more likely to get serology tests at the end, this would elevate their rates of the outcome. It could create selection bias because the denominator is those tested, not total employees in a category (I think, not completely clear), and different groups may have opted for testing at different rates. It creates measurement error (closely related to the confounding) because the test types have different sensitivities and specificities / NPV and PPV.An ideal analysis would adjust for time (by perhaps week) and test type and the interaction (since the shape of the time effect is quite different by test type). There would also be clearer discussion and perhaps adjustment for the issue below in point 2. This may be impossible even with this large data set, but some kind of effort needs to be made. Otherwise the risk factor findings, plausible as they are, are not very robust. I would personally be convinced if these more nuanced analyses showed the same patterns as the overall analysis, even if not as statistically convincing. An overhaul of the analysis of risk factors is needed

We have undertaken additional analyses and expanded our discussion of potential limitations to reflect these points, including clarifying:

1) The asymptomatic staff sampling was undertaken over a relatively short period (23 April onwards) that followed the main peak in patient and symptomatic staff cases at the end of March 2020. As such, the potential for confounding arising from changes in PCR results and seroprevalence over time is more limited than if asymptomatic staff testing had occurred from March 2020 onwards. We have added a new panel to Figure 1 (Figure 1C) to illustrate this, including showing the majority of asymptomatic staff were tested over a 3-week period in May 2020.

2) We now consider the week of testing as a potential predictor in the univariable analysis of risk factors, as suggested by the reviewers. This categorical variable is not selected in the final multivariable model, as it does not improve model fit, i.e. it does not appear to be an important confounder, presumably given the timing of asymptomatic staff testing following the main peak in COVID-19 cases in our hospital. Therefore, the proposed search for interactions with week of testing was not undertaken. We have added discussion of the potential for week of testing-based confounding to our Discussion.

3) The risk factor analysis was undertaken on data obtained from staff attending asymptomatic testing only, and although there were different reasons for testing (symptomatic and asymptomatic staff), only those attending the asymptomatic programme contributed to the analysis – this has been clarified in the manuscript.

4) We discussed as a group of authors whether to present additional separate analyses for our serology and PCR results, we initially settled on presenting a composite endpoint using both results to keep the presentation simpler. However, given the question about test type is raised by the reviewers we have now included a separate analysis of risk factors for SARS-CoV-2 IgG seropositivity only (Supplementary file 1C). This shows very similar results to the main analysis.

2) The other analyses, as well as the risk factor analyses, are confusing in that it is very hard to follow who is being tested with no history of symptoms, past symptoms, future symptoms, etc. Especially confusing is Figure 1 which seems to have only two categories – symptomatic or previously symptomatic, even after asymptomatic testing starts. As stated by another reviewer:In addition, I had a number of questions about what governed the testing and how the distribution of cases changed over time.

Figure 1 A and B provide background context to the asymptomatic staff testing data that follow. The two panels are intended to show the epidemic curve. Therefore, the reason for only including asymptomatic staff who were previously symptomatic is only these staff were able to provide a date of onset of their symptoms. We explain this in the legend. Were we to plot asymptomatic staff diagnoses at the time of their PCR or (in most cases) serological diagnosis this would lead to the visual impression that staff were infected later than they were. However, we accept that the labelling of the x-axis on Figure 1B is potentially unclear and have amended this. In addition, the new Figure 1C shows when asymptomatic staff were actually tested and the proportion of staff testing positive each week.

– For the duration of PCR positivity, what determined whether individuals had repeat tests?

Repeat testing of patients was guided by individual clinician request, in conjunction with the infection and infection prevention and control consult services.

Repeat testing of staff was undertaken in a cohort of staff who attended the asymptomatic testing service during the first week of testing. We also made use of positive tests obtained previously from testing of symptomatic staff.

We have added this information to the manuscript.

– Given the variation in infection control and PPE efforts, how were presumed workplace acquisitions distributed across time? And same with staff roles?

We agree this would be an interesting analysis, however it would require presumed workplace acquisitions in staff to be identified and dated. As most staff were only found to be positive by serology sometime later, the analysis would require a probabilistic reconstruction of when staff likely acquired their infection and the overlaps between staff and between patients and staff.

We have added this limitation to the Discussion, and it is as an area for follow up study.

– Were the symptoms that enabled staff to get tested restricted to fever and cough for the duration of the study period?

Staff and their household contacts were eligible for symptomatic testing if they had a fever (≥ 37.8°C) or a new persistent cough. From 18^th^ May 2020 onwards, testing criteria were expanded to include staff with new onset anosmia in line with national guidance. This clarification has been added to the manuscript.

– For the asymptomatic screening program, did "asymptomatic" mean no fever and cough, or none of the other symptoms associated with COVID-19?

Any staff not meeting the criteria for symptomatic testing, and therefore remaining in work, were eligible for the asymptomatic screening programme. This has been added to the manuscript.

– For PCR testing, were either NP or OP swabs obtained, or both?

Both, this has been clarified in the Materials and methods.

– How incomplete were each of the fields in the reports from participants in the asymptomatic screening program? It seems only 52% reported a date of symptoms. Were all the other fields completed by all participants?

All fields analysed were completed by all participants using a web-based data collection system that required fields to be completed. There were two staff members who were tested in error without registering first, data for these two staff members are not included, but otherwise the data were complete. Count data for all factors is included in Supplementary file 1B.

Not all staff had prior symptoms, where staff did not have symptoms this was explicitly recorded as absence of symptoms, rather than these data fields being missing.

– It seems a number of assays were used. How variable were their test characteristics? Was there any variation in their use over time?

The majority of serological testing was done with both the in-house ELISA to trimeric spike protein and the Abbott nucleoprotein immunoassay. The performance of these two assays has now been described in more detail (Head-to-head benchmark evaluation of the sensitivity and specificity of five immunoassays for SARS-CoV-2 serology on >1500 samples, Lancet Infectious Diseases 2020), which we now reference in the manuscript.

The use of different PCR tests did change over time, and details of the number of successfully completed assay by week are provided in Author response table 1:

Each assay used has regulatory approval and its performance verified in our laboratory prior to use. We have not however undertaken a formal head-to-head comparison of the assays used.

3) Overall, it is also really unclear what outcome is being studied where, exacerbated by the inconsistent headings vs. text (e.g. serologic results heading followed by results on PCR or serologic outcomes).

We have reviewed the headings to ensure these signpost analyses correctly and added additional sentences to Results to make clear which outcomes are being considered.

4) Another limitation of the study is that it was cross-sectional in nature and data on particular exposures may have been subject to recall bias. A potential solution to this would have been to conduct patient and staff screening in parallel from the start of the epidemic but this was not feasible. This should be discussed, at least.

We acknowledge the potential for recall bias when discussing staff recall of exposure without PPE, we have expanded this discussion to cover recall bias more completely.

[Editors' note: further revisions were suggested prior to acceptance, as described below.]

The revision responds adequately to the issues raised, and we have discussed two small changes that would make the paper ready for publication:1) Put the error-bar figures (now supplements) in the main text

We understand this to refer to Figures 4, 5 and 7, which were supplementary in the medRxiv version, but were part of the main text in the version submitted to *eLife*. Please let us know if we have misinterpreted this request as we believe these figures are already part of the main text.

2) Talk about the multivariable model and its meanings. Most of the "adjustments" are in fact blocking causal pathways: for example, ethnicity -> role -> infection risk in the multivariable model is blocked by including role, making the multivariable model less indicative in some cases of likely causal relationships than the univariable ones. Likewise, specialty -> patient-facing (or exposure) -> infection is more meaningfully capturing the specialty effect in univariate (Not conditioning on patient-facing or exposure) than multi. Since many people think multivariable are the "real" effect measures this should be mentioned. The Discussion should convey what the data imply about causality, assuming that unmeasured confounding and other sources of bias are adequately mitigated, this will not be obvious without some explication.

We agree this is important and it is on this basis that we give many of the univariable findings prominence in our Discussion.

There are complex causal relationships represented, and the correct interpretation of the adjusted odds ratios (aORs) presented is important. As such we have expanded the paragraph on personalised risk scores to better help readers interpret the aOR appropriately.

Whether the multivariable or univariable OR for ethnicity better capture causal relationships may depend whether the reader is interested in the influence of ethnicity within the context of existing structural inequalities or in ethnicity having controlled for inequalities as much as is possible in our dataset (via occupation, exposures outside hospital, and specialty). Both are valid questions, with the former answered better by the univariable and the latter by the multivariable estimates.

For specialty, it may be that specialty per se increases risk, e.g. by physical proximity to patients as part of the specialty, e.g. anaesthetists and patient airway exposure, or that it is mediated by particular specialities caring for greater numbers of patients with COVID-19. From our adjusted analysis it appears that the component mediated by specific contact with COVID-19 patients accounts for most of the risk.